# A Multiarmed Bandit Approach for LTE-U/Wi-Fi Coexistence in a Multicell Scenario

**DOI:** 10.3390/s23156718

**Published:** 2023-07-27

**Authors:** Iago Diógenes do Rego, José M. de Castro Neto, Sildolfo F. G. Neto, Pedro M. de Santana, Vicente A. de Sousa, Dario Vieira, Augusto Venâncio Neto

**Affiliations:** 1PPgEEC, Federal University of Rio Grande do Norte, Natal 59078-970, RN, Brazil; martinsee@ufrn.edu.br (J.M.d.C.N.); sildolfo@ufrn.edu.br (S.F.G.N.); pedro.maiadesantana@de.bosch.com (P.M.d.S.); augusto@dimap.ufrn.br (A.V.N.); 2Efrei Research Lab, EFREI Paris, 94800 Villejuif, France; dario.vieira@efrei.fr; 3Vyaire Medical Inc., Cotia 06715-865, SP, Brazil; 4Instituto de Pesquisas Eldorado, Av. Alan Turing, 275-Cidade Universitária, Campinas 13083-898, SP, Brazil; 5Corporate Research, Bosch, Robert-Bosch-Campus 1, 71272 Renningen, Germany

**Keywords:** Wi-Fi, LTE-U, coexistence, reinforcement learning, multiarmed bandit, Q-learning

## Abstract

Recent studies and literature reviews have shown promising results for 3GPP system solutions in unlicensed bands when coexisting with Wi-Fi, either by using the duty cycle (DC) approach or licensed-assisted access (LAA). However, it is widely known that general performance in these coexistence scenarios is dependent on traffic and how the duty cycle is adjusted. Most DC solutions configure their parameters statically, which can result in performance losses when the scenario experiences changes on the offered data. In our previous works, we demonstrated that reinforcement learning (RL) techniques can be used to adjust DC parameters. We showed that a Q-learning (QL) solution that adapts the LTE DC ratio to the transmitted data rate can maximize the Wi-Fi/LTE-Unlicensed (LTE-U) aggregated throughput. In this paper, we extend our previous solution by implementing a simpler and more efficient algorithm based on multiarmed bandit (MAB) theory. We evaluate its performance and compare it with the previous one in different traffic scenarios. The results demonstrate that our new solution offers improved balance in throughput, providing similar results for LTE and Wi-Fi, while still showing a substantial system gain. Moreover, in one of the scenarios, our solution outperforms the previous approach by 6% in system throughput. In terms of user throughput, it achieves more than 100% gain for the users at the 10th percentile of performance, while the old solution only achieves a 10% gain.

## 1. Introduction

With the intense use of mobile phones in everyday life, along with the emergence of the Internet of Things (IoT), mobile data traffic has been experiencing significant growth in recent years. Ericsson has shown [1] that the global mobile data traffic has scaled up 50% between Q3 2019 and Q3 2020 and, by 2026, it will increase by a factor of 4.5, reaching 2226 EB/month. This growth will be driven primarily by smartphone connections, which are expected to account for nearly 95% of the total mobile data traffic.

To tackle this upcoming high demand, mobile operators have adopted different solutions. For example, the use of 5G networks, especially in the range of Millimiter Waves (mmWaves), will provide a significant increase in transmission rate capacity when combined with massive Multiple-Input Multiple-Output (MIMO). However, legacy networks, in their standalone form, need additional improvements, especially when considering the upfront barrier and cost involved in the use of licensed spectrum. With that in mind, mobile operators and standard organizations proposed the use of unlicensed spectra, since unlicensed bands are free from usage costs in terms of purchasing a license of operation.

However, implementing solutions tailored to unlicensed bands requires careful attention to ensure compliance with the regulations of each target region. In order to operate in unlicensed spectrum, certain areas require medium access mechanisms, such as the Carrier Sensing Multiple Access with Collision Avoidance (CSMA/CA), while other regions only need the implementation of a particular mechanism that limits the time a technology can take to control the transmitting medium [2]. The 3GPP proposed three solutions to address these two requirements: (i) LTE-Licensed Assisted Access (LTE-LAA) [3]; (ii) the LTE-Unlicensed (LTE-U) [4], also called LTE-Duty Cycle (LTE-DC); and, (iii) New Radio Unlicensed (NR-U), with either LTE dual-connectivity mode or stand-alone operation [5].

In our previous works [6,7,8,9,10], we have demonstrated how the first two solutions work and how they interact with Wi-Fi, which is currently the most successful technology utilizing the unlicensed spectrum. We also proposed a centralized Q-Learning (QL) solution to increase system throughput in a coexistence multicell scenario with LTE-U and Wi-Fi [10]. The solution provided gain in terms of system and user throughput, and we have identified further improvements for this upcoming work.

In this paper, we extend our previous work by implementing a simpler yet efficient algorithm that achieves higher gain than our previous QL solution and offers improved balance by providing similar throughput for both Wi-Fi and Long Term Evolution (LTE). Our new solution applies an Multi-Armed Bandit (MAB) approach to provide a coexisting framework between Wi-Fi and LTE-U within a multicell environment. We compare the two solutions and demonstrate their performance in each case, indicating which one performs better or worse. We also compare their implementations, highlighting the advantages of our new solution.

This paper is structured as follows. Section 2 presents the related works, followed by Section 3, which presents the evaluation scenario and preliminary results. The problems involved with coexistence in unlicensed bands are introduced in Section 4, and Section 5 presents our proposed MAB solutions. Results are presented in Section 6, and Section 7 concludes with final remarks and future works.

## 2. Related Works

The coexistence of 3GPP systems and Wi-Fi in the unlicensed 5 GHz band began a few years ago when companies and standardization bodies began to outline the corresponding specifications. The goal was to improve user experience by allowing the usage of the unlicensed spectrum while coexisting with Wi-Fi and other technologies.

For example, the LTE-U, originally proposed and led by Qualcomm, is specifically designed for deployments and regions that do not require Listen Before Talk (LBT) and has led to the creation of the LTE-U Forum, which is responsible for defining tests for coexistence scenarios [4,11,12].

3GPP also defined two other solutions in Release 13 [13]: the LTE-LAA and LTE-WLAN Link Aggregation (LWA). LTE-LAA is designed for deployments in Europe and Japan [3,14], while LWA targets deployments where operators already have carrier Wi-Fi networks [15]. However, before specification efforts, 3GPP conducted studies on the feasibility of LTE operating on unlicensed bands. The focus was on fairness, and the criterion was that an LTE network could not impact neighboring Wi-Fi more than another Wi-Fi network.

Qualcomm also developed a proprietary and patented solution, called MuLTEFire, based on 3GPP Releases 13 and 14 for Licensed Assisted Access (LAA/e-LAA) [16,17]. This technology can be deployed without a primary LTE anchor and can coexist or overlap alongside other wireless networks, such as Wi-Fi and MuLTEFire itself. Release 16 of 3GPP systems has standardized 5G NR-U radio access technology as an evolution of the 4G LTE-LAA standards. Similar to MuLTEFire, NR-U could operate in the unlicensed 6 GHz band without any dependency on licensed network operators.

In addition to these, several academic studies have investigated the coexistence between LTE, a technology primarily built for the licensed spectrum, and Wi-Fi, which operates in an unlicensed spectrum. For example, the authors of [18] overview and compare some of the access mechanisms mentioned above, while the authors of [19] show through an experimental analysis that a single LTE parameter can greatly impact WiFi’s performance. Moreover, we want to highlight the studies [6,20,21,22,23], which have made important contributions to the field. We highlight below their most promising remarks:An LTE operator is the best neighbor for a Wi-Fi operator in terms of interference and capacity, even if compared with the coexistence of two Wi-Fi networks.The mechanism for accessing the LTE environment in the unlicensed spectrum must be dynamically adjusted to ensure that no additional configuration is required from the user. The first conclusion is only true if this condition is satisfied.Due to the dynamic aspect of traffic load demand, interference, and usage of different technologies, machine learning techniques are strong candidates for the coexistence problem.

Therefore, numerous papers focus on the features of access mechanisms, and since they define how the technologies can access the spectrum, their parameters must be configured correctly. This helps to ensure that all technologies fairly share the unlicensed spectrum.

For example, in [24], the authors present a comparison between the Duty Cycle (DC) and LBT mechanisms in terms of fairness and optimal throughput performance. They show that if one parameter is properly tuned, the LBT mechanism can achieve a performance limit comparable to the DC. The authors of [11] investigate how the DC adjustment impacts the system throughput on LTE-U, and the authors of [25] present a summary of the problems and solutions involved in achieving fair coexistence between LTE and Wi-Fi. They discuss mechanisms that prevent one technology from harming the other. A notable conclusion is that a Wi-Fi network performs better when coexisting with LTE-LAA than when coexisting with another Wi-Fi.

The coexistence problem is frequently approached as a classic optimization problem, as demonstrated in the papers mapped by the authors of [10]. However, the rest of this section explores the scope of this paper, which includes the use of LTE-U and two Reinforcement Learning (RL) algorithms: QL and MAB.

The authors in [8] propose a framework to maximize the system transmission rate for a controlled interference scenario defined by 3GPP, using the most widespread RL algorithm, Q-Learning. This framework considers the offered data rate to each user and maximizes the system throughput by dynamically choosing the DC parameter. The authors of [26] use QL in a multichannel scenario, with the algorithm dynamically selecting the least populated channel and the DC parameter to improve system performance. In [27], the authors propose a QL algorithm that takes into account the LTE and Wi-Fi queue-length processes as a metric to reflect traffic load. The papers in  [28,29] also employ dynamic DC adjustments with QL similar to the objective of fairness in coexistence, with differences only in the parameters and metrics used. Additionally, recent studies have also explored the use of Deep Q-Learning (Deep QL) as a means to enhance throughput and promote fairness in coexistence scenarios, such as [30,31].

However, modeling QL algorithms requires the definition of several state-action pairs in the target environment. As the number of pairs increases, the algorithm’s complexity also increases, limiting the solution’s flexibility. Thus, modeling a complex and very dynamic environment, such as multicellular wireless scenarios, can be a problem that makes Q-Learning unfeasible as a solution if the requirements for convergence time are too strict. An alternative is to replace Q-Learning with MAB.

One of the first papers to use the MAB theorem to solve wireless communications problems was [32]. The authors investigate solutions for cognitive radio networks in highly dynamic environments, in which the user seeks to take advantage of available frequency bands in a radio spectrum with multiple bands. The authors model the availability of each channel as a Markov chain, and two scenarios are proposed to measure performance: with and without knowledge of the Markov chain. The first models the problem as a competitive multiuser bandit, while the second uses an optimal symmetric strategy to maximize the total user throughput.

In [33,34], the authors propose an MAB for the coexistence of LTE-U and Wi-Fi in the Citizens Broadband Radio Service (CBRS) band at 3.5 GHz. The algorithm provides significant gain by tuning power control and the DC to reduce interference. An MAB algorithm is proposed in [35] for the coexistence of Machine to Machine (M2M) systems with Wi-Fi. The authors model the MAB algorithm to take advantage of white spaces in Wi-Fi transmission, aiming at improving the transmission rate of M2M devices.

In this paper, we present a flexible strategy for adaptive DC selection in a multicellular environment using MAB. Our key contributions are as follows:A simple, adaptive DC selection strategy with better performance and enhanced fairness compared with the Q-Learning-based strategy proposed in [10];A flexible solution for the coexistence that does not require the definition of state-action pairs, allowing a higher number of available DC values to choose from;An adaptive solution that combines low complexity and lower number of parameters to configure;We show that in environments where the rate changes dynamically, the sequential decision-making approach can result in a base model that is applicable for a wide range of reinforcement learning algorithms, whether they are simple or complex;A key comparison of reinforcement learning strategies applied to the adaptive multicell data rate environment for coexistence. To the best of our knowledge, this kind of comparison has not yet been done using the approach presented in this work.

## 3. Evaluation Scenario and Reference Results

In the technical report in [3], 3GPP has defined evaluation scenarios to assess the coexistence between systems that operate on unlicensed spectrum. One of these scenarios is the indoor scenario, which presents the most challenging interference profile, and it has multiple users and multiple cells. This scenario, depicted in Figure 1, has two operators, each with four cells, in a room with dimensions 120 × 50 m and no walls. The distance between access points of different Radio Access Technologies (RATs) is *d*, and between the same RAT is BS_space. In addition to the cells, forty stations are randomly deployed within this rectangular region, with twenty stations from each operator [3].

To model and evaluate the coexistence in this scenario, we use the ns-3 simulator [36]. The ns-3 is a widely used and accepted open-source event discrete simulation tool, which is implemented in C++ and complies with standards defined by IEEE, 3GPP, and Wi-Fi Alliance. To particularly address the coexistence in unlicensed bands, a modified version known as ns3-lbt [37] was developed by the Centre Tecnologic de Telecomunicacions de Catalunya (CTTC), University of Washington, and funded by Wi-Fi Alliance. This modified version implements DC transmission of LTE-U using the approach presented in [38]. According to this approach, LTE-U transmission defines a time window called Almost Blank Subframe (ABS) of 40 ms. As each subframe of an LTE transmission is 1 ms, a bitmask of 40 bits (one bit for the subframe) is defined to control the ON–OFF pattern. This modified version of the simulator can be found on http://code.nsnam.org/laa/ns-3-lbt/file/9529febb7ebc (accessed on 1 January 2023).

In our previous work [10], we presented preliminary results using the same simulator and scenario described above. Our main conclusions, as presented in [10], are summarized below:The DC value that results in maximum system throughput depends on the offered data rate and changes every time that the offered data rates change;If the offered data rate for each operator is unbalanced, the best DC value, resulting in maximum system throughput, is the one that is proportional to the imbalance. For example, if we offer 4 Mbps for LTE-U users and 500 kbps for Wi-Fi users, the optimal DC value is approximately 0.8, because it gives 80% of the channel usage time for LTE-U and 20% for Wi-Fi, matching the demand;A fixed DC value is not enough to achieve maximum throughput in a dynamic system where the data rates are constantly changing. In order to reach maximum throughput in such scenarios, a dynamic algorithm is needed to automatically and dynamically change the DC value based on the system’s current state.Choosing the best DC value for the current state guarantees that the system will always be as close to its maximum throughput as possible, since there is an optimal DC value for each combination of offered data rates;Q-Learning is highly dependent on how the state-action pair is modeled, and this has a great impact on convergence. If the number of state-action pairs is high, the algorithm may not converge before it is time to begin calculating again. On the other hand, if the number of state-action pairs is low, the algorithm may converge to a locally optimal point, or it may not explore all possibilities provided by the environment. Therefore, fine-tuning is necessary when using this approach for optimal decision making.

Based on these conclusions, we developed a framework using the Multi-Armed Bandit problem formulation, and the set of parameters used is adapted from [10], as presented in Table 1. We perform simulation rounds of 20 s for each fixed value of DC and collect the average throughput of each operator. For the proposed framework, two sets of simulations are proposed with different offered data rates. These two sets are detailed in Section 6. These offered data rates are modeled as UDP full buffer, as recommended by 3GPP [3]. Besides, we want to analyze the coexistence when there is always data to transmit. In this case, the transmitter is always trying to access the wireless channel.

Our goal is to maximize system throughput in a coexistence scenario between LTE-U and Wi-Fi. This is performed by automatically choosing the DC value to adapt to dynamic system conditions. First, our problem is formulated as a reinforcement learning model, which is better explained in Section 4. Then, we apply the MAB formulation as a simplification of the RL problem.

## 4. Adaptive Duty Cycle Selection with Reinforcement Learning

Our previous studies have demonstrated that there is a relationship between the DC value and the offered data rate. When the operators’ offered data rate changes, the DC value that maximizes system throughput also changes. In a system where data rates frequently change, using a fixed DC value is not optimal, as it may not lead to maximum throughput. Instead, the system requires an algorithm that automatically selects the best DC value given the current state of the system.

In this paper, we propose a framework that uses MAB to dynamically and autonomously choose the LTE-U DC value in a dynamic scenario where LTE-U coexists with Wi-Fi. The goal of this framework is to maximize system throughput (LTE-U + Wi-Fi) by choosing the best DC value for different combinations of offered data rates. By using this framework, the system is expected to always operate near its maximum achievable throughput, as the framework continuously selects the best DC value based on real-time conditions.

To address the sequential decision-making process of choosing the optimal DC value as the demand changes over time, an Markov Decision Process (MDP) framework is required. From a machine learning perspective, the field of Reinforcement Learning is well-suited to address this behavior [39]. Moreover, dynamic coexistence scenarios require algorithms that can make decisions in real time.

In Reinforcement Learning, the learning process centers around an agent and its interactions with its environment. The agent learns as it applies an action $a_{t}$, at time step *t*, and the environment responds by returning a reward and an observation/state at the next time step t+1. With this reward and state, the agent can find relations between input (actions) and outputs as it works towards a goal. Typically, the goal of a RL problem is to maximize the cumulative reward over time. Once the agent is fully trained, it always chooses the best action on each time step *t* that maximizes the cumulative reward at the end of the learning process. As presented in [39], there are several ways to resolve a reinforcement learning MDP. In this work, we use an MAB approach, as it presents simplifications compared with other methods, resulting in a simple and flexible solution.

In the MAB approach, the agent repeatedly chooses from the same set of actions in each time step. Each action results in a numerical reward based on a probability distribution, which is assumed to be independent and identically distributed (iid) for each action in the set. The goal of the agent is to maximize the expected total reward over a period. If the agent selects one of *k* available actions at each time step, the problem is called a *k*-multiarmed bandit.

In this *k*-MAB, each action corresponds to a reward that follows a probability distribution. Consequently, there is an expected or mean reward associated with each action, referred to as the value of that action, denoted as q(a). Following the notation presented in [39], if the action selected in time step *t* is $A_{t}$, and the corresponding reward is $R_{t}$, the value of $q_{*}\left(a\right)$ for an action *a* is given by
(1)$$q_{*}\left(a\right)≐E\left[R_{t}|A_{t}=a\right],$$
where E denotes the expected value function.

If the value of $q_{*}\left(a\right)$ were known for each action *a*, it would be trivial for the agent to solve the MAB problem by simply selecting either the action with the highest value over the period or time steps. Since these values are not known, we define a function $Q_{t}\left(a\right)$ to approximate $q_{}\left(a\right)$. This function uses an average of the past received rewards for each action at each time step *t* and it is defined as
(2)$$Q_{t}\left(a\right)=\frac{\sum_{i=1}^{t-1}R_{i}\cdot 1_{A_{i}=a}}{\sum_{i=1}^{t-1}1_{A_{i}=a}},$$
where $1_{A_{i}=a}$ is a predicate that is 1 only if $A_{i}=a$ is true. The average is updated only for the selected action at each time step *t*. As we are using an average, it is expected that by the law of large numbers, the value of $Q_{t}\left(a\right)$ will converge to $q_{*}\left(a\right)$ as the denominator tends to infinity.

The best action to be selected in time step *t* is the action with the highest estimated value following a greedy approach as
(3)$$A_{t}≐\underset{a}{argmax}\;Q_{t}\left(a\right),$$
exploiting the current agent’s knowledge about the environment. However, if the agent only uses a greedy approach, it might get stuck in a suboptimal action. For that reason, the agent should also explore by selecting actions at random to discover new options. That exploration versus exploitation dilemma is a well-known concept in the reinforcement learning field, as it applies to nearly every problem.

One way to solve it is to define a variable ϵ that represents the probability of randomly selecting an option from the set of actions, each with equal probability. This strategy is known as ϵ-greedy. At each time step *t*, the agent has a probability of 1−ϵ to select the greedy action, prioritizing exploitation, and a probability of ϵ to select an action randomly, promoting exploration. This approach tries to balance between exploring different actions and exploiting the actions that have shown good results.

## 5. MAB-ADC: Multiarmed Bandit for Adaptive Duty Cycle Selection

This section introduces the proposed Multi-armed Bandit for Adaptive Duty Cycle Selection (MAB-ADC) framework, which utilizes MAB for adaptive DC selection. The algorithm embedded in the LTE-U cell serves as the agent. Unlike conventional RL algorithms, MAB does not define any states. Instead, there is only one state, as the $Q_{t}\left(a\right)$ depends only on the action itself.

The whole framework is presented below, including each variable from the MAB theory mapped to its corresponding counterpart in the coexistence problem:The actions that the agent can choose from are the duty cycle values in the set A={0.1, 0.2, 0.3, 0.4, 0.5, 0.6, 0.7, 0.8}. Each value represents how much channel air time, during the ABS time, is given to each access technology. For example, a DC value of 0.4 means that 40% of the ABS time is reserved for LTE-U and 60% is reserved for Wi-Fi;The reward is defined as the aggregated throughput: $T_{x_{lte}}+T_{x_{wifi}}$, where $T_{x_{lte}}$ is the LTE-U operator throughput over the ABS time, and $T_{x_{wifi}}$ represents the throughput of the Wi-Fi operator;The decision-making process happens for each duty cycle duration of 40 ms (ABS time);To balance the exploration vs. exploitation trade-off in the ϵ-greedy algorithm, the value of ϵ is initially defined as 0.3, which corresponds to a 30% exploring probability. Then, this value decays with a factor of 1.015. This configuration ensures more exploration at the beginning and more exploitation after learning the best action. Besides, these values result in an exploring probability of 1.5% after 200 random selections.

Algorithm 1 presents the pseudoalgorithm for the MAB-ADC. The variable N(a) is a vector that keeps track of how many times each action has been chosen up to the current time. The updating equation for Q(a) is modified to its incremental form, as opposed to Equation (Equation 1). The main advantage of using this incremental form is that it only needs to store Q(a) and N(a), minimizing the memory requirement.

**Algorithm 1:** MAB-ADC: MAB application for adaptive DC selection

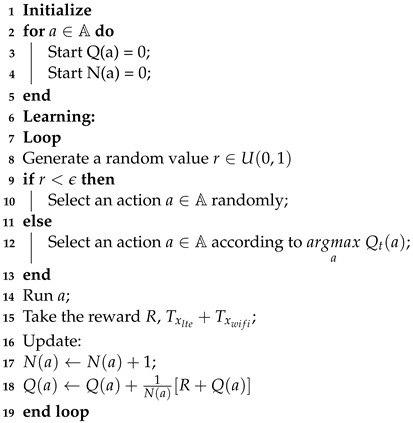



To further enhance our proposed framework based on MAB, we add two operation modes: the independent and the centralized coordinated. In the independent mode, each LTE-U cell has its own embedded MAB-ADC, and the DC selection is independent among cells, i.e., each cell calculates its own ϵ, selects an action, and updates the *Q*(*a*) independently. The centralized coordinated mode has a central node where the proposed MAB-ADC is running and information from all LTE-U cells is exchanged. This central node calculates the ϵ, updates the variables, and selects the next actions. After defining the next action, that node sends the DC value to all connected LTE-U cells, and all of them set that same DC value until the central node defines a new one at the next ABS time.

## 6. Performance Evaluation of MAB-ADC

We evaluate the impact of our proposed coexistence framework in two sets of simulations. The first set comprises a short simulation to assess the framework’s ability to switch between different DC values in response to a change in the environment. The second set is a longer simulation that considers different offered data rates.

It has 40 s of duration, and the offered data rate changes after 20 s;The offered data rate is initially 4 Mbps for LTE-U users and 500 kbps for Wi-Fi users. At 20 s, the data rates are inverted. This approach comprises two different unbalanced scenarios regarding the system’s demand. When LTE-U users have a much higher offered data rate, the optimal DC value is the one that gives more channel time to LTE-U, that is, a DC value closer to 1. However, if Wi-Fi users have a higher offered data rate, the optimal DC value is the one that gives more channel time to Wi-Fi, closer to 0;The algorithms always operate at the end of the current ABS time (40 ms windows). At that time, they select the DC value for the next ABS interval;For comparison, we use the Q-Learning framework proposed in [10]. It has the same centralized and coordinated operation as one of our proposed MAB operation modes. The QL is defined with four actions {0.2, 0.4, 0.6, 0.8} and four states. Since QL also requires the state’s definition, we established only four actions. If the number of actions were to be increased, the convergence time would be affected.

For easier comparison, we have included the results of our previous Q-Learning solution in the same figures as the new proposed framework. Whenever we mention QL results, they belong to our previous work.

Figure 2 shows a bar graph of the operator’s throughput as a function of the fixed Duty Cycle, the previous Q-Learning solution, and the proposed solutions, namely the independent MAB and the coordinated MAB. Comparing the aggregated throughput with the best fixed DC value of 0.5, the MAB solutions present a gain of 9 Mbps (15% gain) and 12 Mbps (20% gain) for the independent and coordinated modes, respectively. However, Q-Learning provides the highest gain, nearly 15 Mbps, which corresponds to a 25% improvement.

Both coordinated solutions, MAB and QL, were able to achieve maximum system throughput. However, the LTE operator suffers a 20% loss, with the Wi-Fi operator showing a significantly higher throughput compared with LTE. The independent MAB, on the other hand, was able to provide positive gain for both operators, while still showing a substantial gain in system throughput. It improves the LTE by over 30% and Wi-Fi by approximately 2%, providing similar throughput for both operators. This behavior may be caused by the different degrees of freedom. In the coordinated solution, the base stations must use the same DC value determined by the proposed framework. On the other hand, in the independent solution, each base station has its own DC value, providing a higher degree of freedom and thus allowing different combinations of DC values.

The histogram in Figure 3 shows that during the simulation for coordinated MAB, the most frequently selected DC values were 0.3 and 0.7. This observation aligns well with the expected behavior, since a simulation that is divided into two very different parts should also present two optimal values.

For the first 20 s, the offered data rate is 4 Mbps for LTE and 500 kbps for Wi-Fi. Given this configuration, the best DC value in terms of throughput is expected to favor LTE when allocating transmission time. Consequently, the optimal value found by the coordinated MAB was 0.7, which gives 30% of the time window for Wi-Fi and 70% for LTE. From 20 s on, the offered data rates are inverted. Hence, the best DC value in terms of throughput should now favor Wi-Fi when allocating transmission time. For that case, the algorithm selected 0.3, which gives 70% of the window time to Wi-Fi and 30% to LTE.

It is also important to note that the DC value of 0.7 is selected fewer times than 0.3, even though the simulation is evenly split into two parts. This behavior is due to the simulation software used. The ns-3 simulator allocates an initial time to configure the servers and start the services. As a result, in the first part of our simulation, when 0.7 is the optimal DC value, the proposed framework has less time to act due to this warm-up period. However, this behavior does not impact the performance, as the results demonstrate that the algorithms effectively select the optimal DC value in each half of the simulation.

The second set of simulations complements our conclusions about the proposed solutions in a more complex scenario. The data rate change is more dynamic and the simulation is longer. This new scenario is defined as follows:The simulation has a duration of 250 s, and the offered data rate changes at specific timestamps throughout the simulation;The offered data rate is uniformly sampled from the set $T_{oferred}$ = { 500 kbps, 1 Mbps, 2 Mbps, 4 Mbps};The specific timestamps where the offered data rate changes occur are uniformly sampled from an interval between 10 and 15 s. These timestamps are added to the current simulation time until it reaches the 250 s limit. As a result, during the 250 s simulation, there will be from 16 to 25 changes in the offered data rate (calculated by dividing the total duration by the minimum and maximum intervals);The algorithms always operate at the end of the current ABS time (40 ms windows). At that time, they select the DC value for the next ABS interval;To ensure statistical confidence, the 250 s simulation was repeated 100 times. In each simulation run (snapshot), the stations are uniformly distributed across the multicell scenario, resulting in different interference profiles, since the stations are in different positions at each snapshot;For comparison, we also use the Q-Learning framework defined in [10]. It has the same centralized and coordinated operations as one of our proposed MAB operation modes.

Figure 4 presents a bar graph of the operator’s throughput for the second set of simulations. The set of fixed DC values is compared with the proposed solutions and the reference Q-Learning approach. The independent MAB has the highest gain, outperforming the best fixed DC value of 0.6 by approximately 3.5 Mbps. However, the gain achieved by the coordinated MAB and Q-Learning is almost the same as the gain shown in the independent MAB.

Regarding the operators, their throughput is more balanced with the independent MAB and the Q-Learning solution and more unbalanced with the coordinated MAB. Additionally, compared with the first set of simulations, all solutions achieve smaller gain. This strongly indicates a dependent relationship between the gain and the difference in the offered data rates. When the offered data rates are more unbalanced, as in the first set of simulations, the solutions present higher gain. On the other hand, if the offered data rates are more equalized, as in the second set of simulations with randomly selected data rates, the gain is smaller.

Despite the solutions being designed and modeled to maximize system throughput, they also lead to better user throughput. Figure 5 shows how each solution performs compared with the best fixed DC value of 0.6, specifically in terms of the 10th percentile. The coordinated MAB, although it provides gain for Wi-Fi users, does come at the cost of a noticeable loss for LTE-U users. The same behavior is shown for the Q-Learning approach, but the ratio loss/gain is more balanced. In contrast, the independent MAB demonstrates a gain of more than 100% compared with the fixed DC value of 0.6, with only a loss of less than 10% for Wi-Fi users.

Figure 6 sketches an overview of the scenario using a Cumulative Distribution Function (CDF) that compares the discussed solutions. The CDF of the user’s throughput is useful to visualize the performance of different groups of users and to assess how each solution impacts each group. Since in Figure 4 the value 0.6 is the fixed DC value that results in higher throughput, we compare its results with the coordinated MAB and the independent MAB for both LTE-U and Wi-Fi users.

According to the CDF, the independent MAB shows a throughput gain for more than 50% of the worst LTE-U users, while Wi-Fi raises almost no loss compared with the reference curve of a fixed 0.6 DC value. The coordinated MAB, on the other hand, shows a significant gain for all Wi-Fi users, but that gain comes at the expense of the LTE-U users, particularly the worst-performing ones in the coordinated MAB approach. These results are in accordance with the discussions from Figure 2 and Figure 4. The independent MAB solution is better at providing similar throughput for both operators.

## 7. Final Comments and Future Investigation

In this study, we investigated the use of two Reinforcement Learning algorithms in a multicell scenario involving the coexistence of LTE-U and Wi-Fi networks in an unlicensed band. We developed two approaches, referred to as independent and coordinated MAB, and compared them with a framework based on the well-known Q-Learning algorithm.

Overall, two main reasons indicate that the independent MAB is the best solution: (i) across our two sets of simulations, this solution improves the aggregated throughput and provides the best fairness among operators; (ii) the independent MAB delivers similar throughput gain compared with Q-Learning, but its implementation involves fewer parameters and more simple calculations, resulting in lower complexity. For example, since there is no need to define states, it allows for a bigger set of possible actions without compromising convergence.

Finally, despite the solutions being modeled to improve the aggregated throughput, they also enhance user throughput. The independent MAB demonstrated less than 10% of throughput loss for Wi-Fi users, while LTE-U users with the worst performance experienced a gain of over 100%.

Our further investigations include the coexistence analyses of the proposed solutions with LTE-LAA and NR-U, as well as the conception of RL solutions to Licensed Assisted Access (LAA) (LTE-LAA and NR-U). We also consider investigating Wi-Fi parameters that could also be dynamically configured to improve even further the flexibility of our solution.

## Figures and Tables

**Figure 1 sensors-23-06718-f001:**
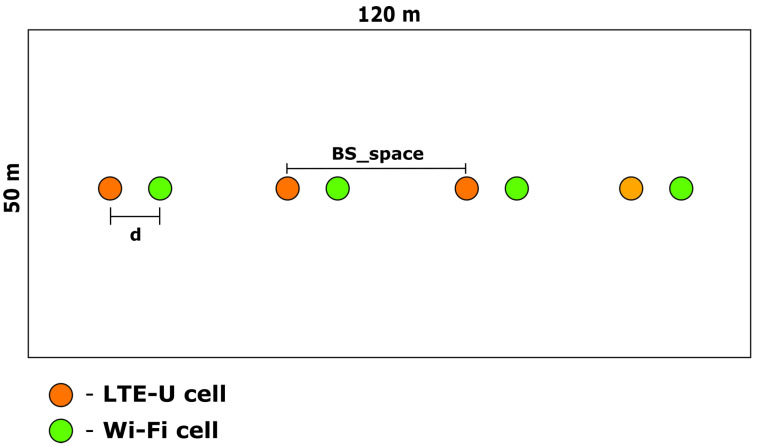
Indoor coexistence scenario proposed by the 3GPP [3].

**Figure 2 sensors-23-06718-f002:**
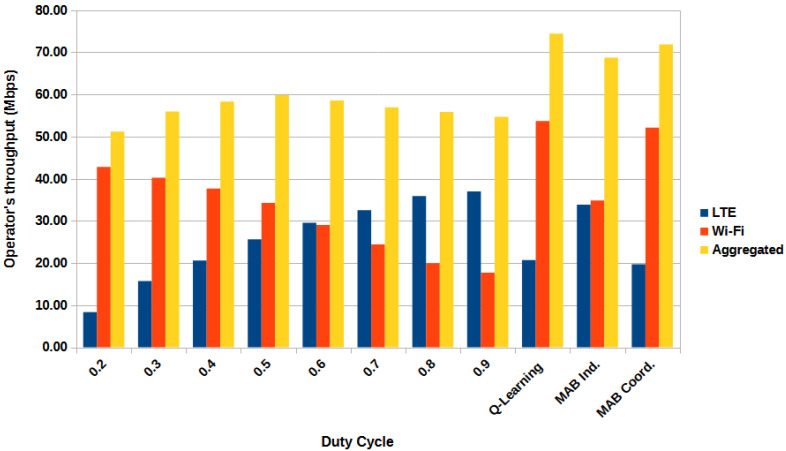
Simulation 1: Operator’s throughput vs. fixed DC and proposed solutions.

**Figure 3 sensors-23-06718-f003:**
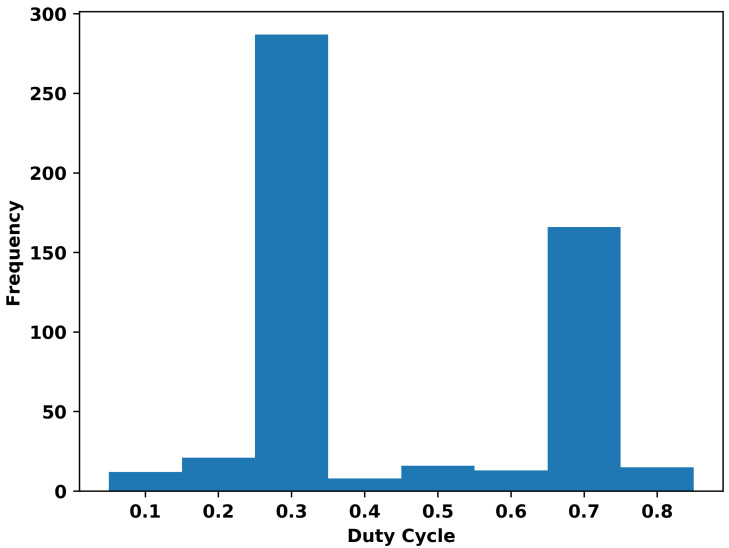
Histogram of the chosen DC value for the first set of simulations of coordinated MAB.

**Figure 4 sensors-23-06718-f004:**
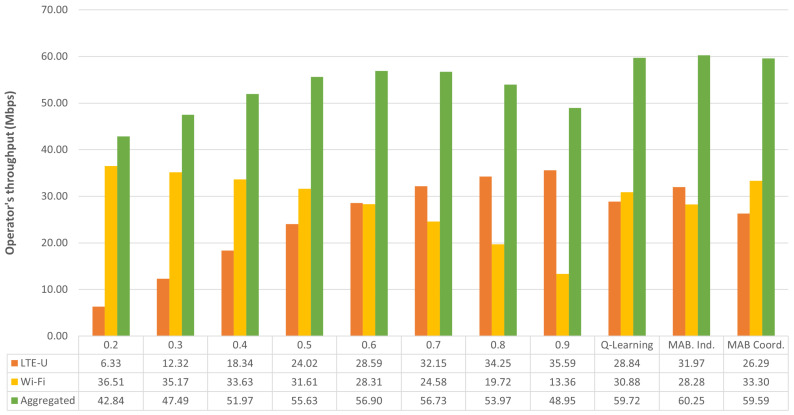
Operator’s throughput for the second analyzed simulation.

**Figure 5 sensors-23-06718-f005:**
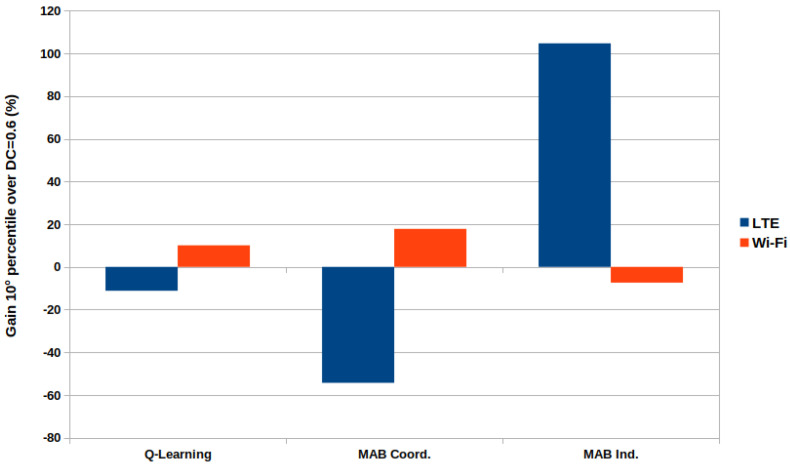
Relative gain of 10th percentile user’s throughput.

**Figure 6 sensors-23-06718-f006:**
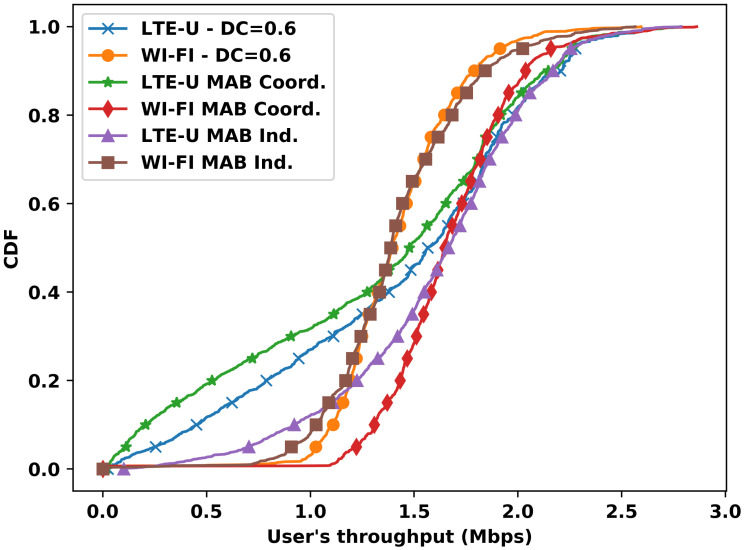
CDF of the user throughput for the MAB solutions and a fixed DC value of 0.6.

**Table 1 sensors-23-06718-t001:** Simulation parameters (adapted from [10]).

Wi-Fi parameter (802.11n-HT PHY/MAC)	
Bandwidth	20 MHz
CCA-Energy Detection threshold	−62 dBm
CCA-Carrier sense threshold	−82 dBm
Bit Error Rate (BER) target	$10^{-6}$
**LTE parameters**	
Bandwidth	20 MHz
Packet scheduler	Proportional fair
ABS pattern duration	40 ms
Duty Cycle (DC) values	{0.2, 0.3, 0.4, 0.5, 0.6, 0.7, 0.8, 0.9}
**Common parameters**	
Tx power	−18 dBm
Traffic model	UDP full buffer
Mobility	Constant position
**Scenario**	
*d* (distance between APs of different RATs)	5 m
bs_space (distance between APs of same RAT)	25 m
Number of LTE-U APs	4
Number of LTE-U stations	20
Number of Wi-Fi APs	4
Number of Wi-Fi stations	20
Path loss and Shadowing	ITU InH
Cell selection criteria	For Wi-Fi, AP with strongest RSS.
	For LTE-U, cell with the strongest RSRP.
UDPRate	{0.5, 1, 2 Mbps, 4 Mbps}.

## Data Availability

Not applicable.

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
