# Peer review of "A Multiarmed Bandit Approach for LTE-U/Wi-Fi Coexistence in a Multicell Scenario"

_sensors, 2023, doi:10.3390/s23156718_

Round 1
Reviewer 1 Report
The authors presented an extension work of their previous solution by implementing a simpler 9 and more efficient algorithm based on the Multi-armed bandit theory. The paper is well organized and has a limited contribution, however, I have some comments and suggestions to improve the paper further as follows:
- In the abstract, the authors should present a comparison of results between their previous work and the current work (i.e. percentage of improvement)
- In the introduction, the authors mentioned that "In this paper, we extend our previous work by implementing a simpler yet efficient algorithm that is capable of achieving equal or higher gain than our previous Q-Learning 46 solution" If this work achieves similar to the previous one, so what is the contribution?
- Most of the related works are outdated and you need to add more recent works.
- I don't see any comparison between the proposed work and previous work as the authors mentioned there is an improvement compared to the previous work.
- Please complete the caption description of Figure 6. CDF.
The article need a proofreading
Author Response
Thank you for all your comments and suggestions. Please refer to the PDF file for our detailed response.

Reviewer 2 Report
There is an unfinished sentence in Page 5 "For example, if we offer 4 Mbps for LTE-U users and 500 kbps for Wi-Fi 175 users, the optimal DC value is approximately 0.8, because it gives 80% of the channel 176 usage time for"
Figure 3 should have units in Frequency, as you did in other figures =)
For performance evaluation you set initially 4Mbps and 500kbps to LTE and WiFi respectively. There is no discussion or motivation on why is that. Also since simulations are free the contrary could also have been simulated, to have a better view of the behavior.
It would be interesting to have simulated a WiFi only scenario to know what would have been the maximum achievable throughput achieved and then compare with the one with the two technologies.
The same way you are proposing an orchestrated/centralized LTE DC selector, wouldn't it be possible to introduce the WiFi APs in the equation so that they also got reconfigured depending on the LTE decision or in concordance with it. I'm thinking in WLAN channel selection, for instance. I understand that it is a multi-operator scenario but for the same operator that would be possible.
In that line, I'm missing a comparison depending the number of operators. Since one operator in the centralized use case should be able to take big advantage of the deployment.
There is a lack of references newer than 2020. That can only mean two things, either the State of the art hasn't been updated properly, either it is a field of low interest (I don't think so). So an update on the State of the art is really needed.
I only found a problem in figure 4 where there is the Brasilian term Agregado instead of aggregated. Apart from that the English is fine.
Author Response

(The authors gave the same response as above.)

Reviewer 3 Report
It can be seen from list of sourses that the authors are well versed in the problem of the coexistence of two wireless technologies on an unlicensed band and therefore the paper is written clearly, in good language and easy to read. It is positive that in the paper, when solving a fairly well-known problem, with a large number of well-known published works, the multi-armed bandit method is used.
Despite the generally positive evaluation of the paper, there are several comments that authors are advised to pay attention to when preparing the manuscript for publication.
1. It is not customary to use the abbreviation MAB in the title of an article. It is rather rare, usually written in full.
2. If the MAB is placed in the head of the paper, then this method can be considered to provide advantages over other approaches. However, there is no such comparison in the numerical analysis.
3. In Table 1, the variable "d" is written without explaining what it means. There are also a number of similar typos that should be corrected when proofreading the work.
4. From the introduction it can be understood that the main result of the work is the improvement by the authors of their own approach from the previous works. It is necessary to clearly articulate what exactly has been done in the paper in terms of technology, which was proposed by 3GPP in 2015.
Author Response

(The authors gave the same response as above.)

Round 2
Reviewer 1 Report
Thanks for addressing my comments, but there are several minor corrections that you need to address in your paper to improve it further:
- Please check all short definitions and some of them are defined several times such as MAB.
- Cognitive Radio Network is a new technique that utilizes available unlicensed spectrum bands due to limited numbers of fixed licensed spectrum bands. which can be used also for LTE-U and WIFI. You may also need to cite the following paper "Efficient handoff spectrum scheme using fuzzy decision making in cognitive radio system"
- The labeling in figures 2,5 and 5 are not explained in the text and also not defined: MAB Coord./ MAB ind. Additionally, you need to differentiate between your previous work and current work.
- Need more discussion on Figure 6.
Author Response
Thanks again for the comments and suggestions! Please see the attachment.
